# Estimation of Pressure Pain in the Lower Limbs Using Electrodermal Activity, Tissue Oxygen Saturation, and Heart Rate Variability

**DOI:** 10.3390/s25030680

**Published:** 2025-01-23

**Authors:** Youngho Kim, Seonggeon Pyo, Seunghee Lee, Changeon Park, Sunghyuk Song

**Affiliations:** 1Department of Biomedical Engineering, Yonsei University, Wonju 26493, Republic of Korea; psgzzing@yonsei.ac.kr (S.P.); fhrm502@yonsei.ac.kr (S.L.);; 2Department of Robotics & Mechatronics, Korea Institute of Machinery & Materials, Daejeon 34103, Republic of Korea

**Keywords:** pain, quantification, electrodermal activity (EDA), tissue oxygen saturation (StO_2_), heart rate variability (HRV), machine learning, pressure intensity, pain level

## Abstract

Quantification of pain or discomfort induced by pressure is essential for understanding human responses to physical stimuli and improving user interfaces. Pain research has been conducted to investigate physiological signals associated with discomfort and pain perception. This study analyzed changes in electrodermal activity (EDA), tissue oxygen saturation (StO_2_), heart rate variability (HRV), and Visual Analog Scale (VAS) under pressures of 10, 20, and 30 kPa applied for 3 min to the thigh, knee, and calf in a seated position. Twenty participants were tested, and relationships between biosignals, pressure intensity, and pain levels were evaluated using Friedman tests and post-hoc analyses. Multiple linear regression models were used to predict VAS and pressure, and five machine learning models (SVM, Logistic Regression, Random Forest, MLP, KNN) were applied to classify pain levels (no pain: VAS 0, low: VAS 1–3, moderate: VAS 4–6, high: VAS 7–10) and pressure intensity. The results showed that higher pressure intensity and pain levels affected sympathetic nervous system responses and tissue oxygen saturation. Most EDA features and StO_2_ significantly changed according to pressure intensity and pain levels, while NN interval and HF among HRV features showed significant differences based on pressure intensity or pain level. Regression analysis combining biosignal features achieved a maximum R^2^ of 0.668 in predicting VAS and pressure intensity. The four-level classification model reached an accuracy of 88.2% for pain levels and 81.3% for pressure intensity. These results demonstrated the potential of EDA, StO_2_, HRV signals, and combinations of biosignal features for pain quantification and prediction.

## 1. Introduction

Research on discomfort and pain response remains ongoing, but methods for objectively and quantitatively measuring comfort between machines and humans are still lacking [1]. Pain varies among individuals due to factors such as gender, cultural background, and other influences, making consistent measurement challenging. The International Association for the Study of Pain (IASP) defines pain as “an unpleasant sensory and emotional experience associated with actual or potential tissue damage”, and common scales include the Visual Analog Scale (VAS), Numerical Rating Scale (NRS), and Likert scale [2,3]. The VAS is widely used in the medical field for its simplicity and effectiveness in evaluating patients’ conditions, despite relying on subjective responses. However, the reliability of the VAS may decrease if subjects struggle to distinguish between the 0–10 levels, and subjective assessments may lead to the misuse of prescriptions. The CDC estimates that around $500 million is lost annually due to this issue [4,5,6]. Thus, quantitative and objective pain assessment methods, while ensuring the reliability of subjective reports, are necessary [7,8,9].

### 1.1. Electrodermal Activity (EDA)

Many studies have quantified discomfort and pain responses using electrodermal activity (EDA) [10,11,12,13]. EDA is regarded as a reliable method for assessing sympathetic responses to external stimuli, measuring variations in skin conductance driven by sweat gland activity in the fingers, palms, and soles. EDA signals are separated into phasic (Skin conductance response, SCR) and tonic (Skin conductance level, SCL) components using a high-pass filter (0.05 Hz) for analysis [14,15,16]. SCR events are calculated from the phasic signal and are considered significant EDA responses if the onset-to-peak value exceeds 0.05 μS. Kim et al. [16]. showed that as the intensity of pressure increases, the SCR amplitude increases linearly, and significant changes in measures were observed in response to pressure algometry. However, consistent evaluation of EDA is challenging due to the large variability in responses among subjects. Posada-Quintero et al. [17,18] proposed a time-varying index of electrodermal activity (TVsymp) and a modified time-varying index of electrodermal activity (MTVsymp), obtained through variable frequency demodulation, to overcome these limitations. Kong et al. [19] confirmed that TVsymp and MTVsymp could effectively reflect thermal pain responses. Furthermore, Kong et al. [9] showed that these methods could classify the presence or absence of pain with an accuracy of up to 84.2%, while Posada-Quintero et al. [8] demonstrated that they could classify pain intensity and the three levels of pain with an accuracy of 68.9%.

### 1.2. Heart Rate Variablity (HRV)

HRV is a method that measures the variation in the time intervals between heartbeats, offering a non-invasive way to assess autonomic nervous system (SNS and PNS) responses, and it serves as a reliable indicator of the autonomic nervous system’s responsiveness to nociceptive stimuli [20,21,22]. Changes in HRV features (such as NN interval, RMSSD, SDNN, HF, LF, etc.) have been reported in response to various pain stimuli, including temperature, pressure, and electrical stimulation. For example, RMSSD increased when pressure was applied to the calf’s gastrocnemius muscle [23], and HF significantly decreased when heat pain was induced in the forearm [24]. However, most studies have analyzed HRV changes based on the presence or absence of pain, and few have examined HRV differences based on pain intensity. Pain and HRV changes vary among individuals and are influenced by factors such as ethnicity, gender, age, breathing patterns, and emotional state [22,25,26]. Generally, pain leads to an increase in sympathetic nervous activity and a decrease in parasympathetic activity, with HRV changes observed in response to “clear pain” (e.g., electrical or temperature stimuli).

### 1.3. Tissue Oxygen Saturation (StO_2_)

Pressure can cause local deformation of the body, leading to impaired blood circulation and microvascular occlusion [27]. Pressure-induced stimuli are commonly encountered at the interface between machines and humans, and these stimuli are more prevalent in daily life than thermal or electrical stimuli. They often arise from factors such as body weight or carrying a heavy bag, and when sustained, can lead to discomfort, pain, or soft tissue damage due to ischemia. Pressure applied in a fixed manner to the human body deserves particular attention due to its potential effects. Linnengerg et al. [28] confirmed a decrease of approximately 4.1% in oxygen saturation per minute at the interface between the machines and the body. Kermavna et al. [29] measured the impact of pressure applied to the thigh and calf on the discomfort and pain experienced by subjects using near-infrared spectroscopy (NIRS). Kermavna’s study aimed to quantify changes in tissue oxygen saturation (StO_2_) using the ratio of oxyhemoglobin (OxyHb) to deoxyhemoglobin (DeoxyHb). There is a lack of research measuring changes in StO_2_ in response to pressure stimuli in the knee region, while Kermavna’s study [30] measured discomfort and pain by applying a cuff around the knee. Nam et al. [31] confirmed a decrease in StO_2_ over 2, 5, and 10 min of pressure application, and Kim et al. [32] showed that the decrease in StO_2_ increased as pressure intensity increased at the thigh. These results showed the potential of using StO_2_ to quantify discomfort and pain.

### 1.4. The Purpose of the Study

This study aims to quantify pain and discomfort induced by pressure by simultaneously analyzing sympathetic nervous responses and blood flow changes. Understanding how pain is perceived and its intensity evaluated is critical for comprehending human responses to external stimuli, predicting and classifying pain effectively, and improving user interfaces. This study examines changes in biosignals, including EDA, StO_2_, and ECG, in response to pressure applied to the thigh, knee, and calf. The correlation between these physiological signals and pain levels (VAS) is analyzed to explore their potential for predicting and classifying pain or discomfort. Statistical analysis, regression, and classification models were used to investigate relationships between biosignals and pain, as well as to assess differences based on the thigh, knee, and calf. This approach provides a foundation for quantifying discomfort and pain solely based on biosignals and could play a key role in advancing pain assessment and improving user comfort.

## 2. Materials and Methods

### 2.1. Experiments

A total of 20 healthy Asian adults (10 females, 10 males; 22.3 ± 2.1 years, 167.5 ± 5.7 cm, 65.5 ± 14.6 kg) participated in this study. All subjects were fully informed of the contents of the experiment and provided written consent for this study. The experimental procedures were approved by the Institutional Review Board of Yonsei University Mirae (1041849-202402-BM-032-03). The following subjects were excluded: (1) BMI over 40; (2) pathological conditions in the musculoskeletal or nervous systems of the lower limbs; (3) current or past use of psychotropic drugs affecting the sympathetic nervous system; Subjects were instructed to refrain from consuming caffeine and alcohol for 24 h prior to the experiment, as well as fasting for 3 h before the experiment. Discomfort and pain were measured using the Visual Analog Scale (VAS), which ranges from 0 to 10. A score of 0 shows no pain or discomfort, while 10 represents the maximum pain experienced by the participant. VAS was categorized as follows [8]: 0 indicates “no pain”, 1–3 indicates “noticeable but does not interfere with daily activities (Low)”, 4–6 indicates “interferes with daily activities or causes discomfort (Moderate)”, and 7–10 indicates “inability to perform daily activities and significant pain (High)”. A guideline was provided to assist subjects in independently assessing their pain level.

### 2.2. Equipments

Pneumatic Cuffs and Pressure-Applying Devices

A Pneumatic cuff (no-pinch single cuff, 86 cm × 10 cm (DTC-S07), DS Maref, Gunpo, Republic of Korea) was used to apply pressure. The pressure stimulus was adjusted using a pressure device (MoorVMS-PRES, Moor Instruments Ltd., Axminster, UK).

EDA

Ag and AgCl hydrogel electrodes were used to measure EDA. The electrodes were attached to the hypothenar and thenar of the left hand (Figure 1a). EDA signals were measured at a sampling rate of 1000 Hz through a wired connection to a GSR100C (Biopac Systems Inc., Goleta, CA, USA), MP150 (Biopac Systems Inc., Goleta, CA, USA), and AcqKnowledge 5.0 Software (Biopac Systems Inc., Goleta, CA, USA).

ECG

ECG was measured using Ag and AgCl hydrogel electrodes, which were attached to the right wrist (−), right ankle (Ground), and left ankle (+) (Figure 1b). ECG signals were measured at a sampling rate of 1000 Hz through a wired connection to an ECG100C (Biopac Systems Inc., Goleta, CA, USA) and AcqKnowledge 5.0 Software (Biopac Systems Inc., Goleta, CA, USA).

Near-Infrared Spectroscopy (NIRS).

Near-infrared spectroscopy (MoorVMS-NIRS, Moor Instruments Ltd., Axminster, UK) was placed under the pneumatic cuff and attached to the thigh, knee, and calf (Table 1, Figure 2) to observe changes in blood flow. Using infrared light in the 750–850 nm range, the sensor measured the concentrations of oxygenated hemoglobin (OxyHb) and deoxygenated hemoglobin (DeoxyHb) in the local tissue with a sampling rate of 40 Hz. The tissue oxygen saturation (StO_2_) was calculated based on the ratio of OxyHb to DeoxyHb (Equation (1)). Since the NIRS sensor can be affected by skin tone and conditions of attachment sites [33,34,35], measurements were conducted with subjects of similar skin tones, and the sensor was attached to areas free from pigmentation or lesions.(1)$$Tissue\;oxygen\;Saturation\left(StO_{2}\right)=\frac{OxyHb}{OxyHb+DeoxyHb}\times 100[\%]$$

### 2.3. Method

The experiment was conducted in a silent space, where the subjects were seated comfortably. The average laboratory temperature was 24 ± 1.06 °C. Subjects wore shorts and were barefoot during the experiment to facilitate the application of sensors and pneumatic cuffs. The pneumatic cuffs were worn in a form that wrapped around the body, as shown in Figure 2, and applied circumferential pressure to the subjects. Subjects waited for 5 min for stabilization before the experiment began. Subjects were asked to minimize movement and avoid head or chin motion, as well as speaking during the experiment. The subjects’ movements were monitored through a camera, and the experiment was stopped if any large movements occurred. Each subject participated in three experiments: on the first day, pressure was applied to the thigh; on the second day, to the knee; and on the last day, to the calf. The experiment consisted of 9 trials, with pressures of 10, 20, and 30 kPa applied three times each. The state with no applied pressure, defined as 0 kPa (baseline), was measured while only the cuffs were worn. The 0 kPa measurement was taken before applying 10 kPa pressure. Each pressure intensity was applied for 3 min, with the order of 10, 20, and 30 kPa repeated. VAS measurements were obtained during the recovery time. The experimental procedure is shown in Figure 3. Sympathetic nervous system responses, such as visual, auditory, or cognitive reactions, may interfere with those caused by pressure, making physiological signals under prolonged pressure difficult to interpret. All sensors were fixed with medical tape to minimize movement noise, and the chair height was adjusted so that the subjects’ feet could reach the floor to prevent suspension and minimize potential movements.

### 2.4. Data Processing

The EDA, StO_2_, and HRV features were extracted from this 3 min period. These features were then analyzed in relation to pressure intensity and pain levels. These features were applied to regression and classification models to predict pain levels and pressure intensity.

#### 2.4.1. EDA Signal Processing

EDA signals were analyzed using Python 3.8 and the Neurokit2 EDA library [36], with the data divided into four components. A 4th-order Butterworth low-pass filter with a cutoff frequency of 3 Hz was applied for preprocessing. Noise-affected segments were identified using BIOPAC AcqKnowledge 5.0 and then removed and replaced through linear interpolation [37]. Data from the 3 min pressure application period were separated into Phasic and Tonic components using a 0.05 Hz high-pass filter. SCR events were defined as those exceeding 0.05 µS within the Phasic component, and the number of events was calculated as SCR counts. TVSymp and MTVSymp were calculated using frequency-variable demodulation [9,19] in the 0.08–0.24 Hz range of the EDA signal, as described in previous studies. The maximum values, averages, and standard deviations of the Phasic, Tonic, SCR events, and MTVSymp were extracted as features (Figure 4).

#### 2.4.2. HRV Signal Processing

HRV signals were extracted based on the ECG signals from the 3 min interval during which pressure was applied. The extracted signals were processed using the Neurokit2 HRV library [36,38] in a Python 3.8 environment to calculate time-domain features, including MeanNN, MaxNN, MinNN, SDNN, RMSSD, and pNN50 and pNN20, as well as frequency-domain components such as HF, LF, HFn, LFn, and HFLF. A 5th-order Butterworth high-pass filter with a cutoff frequency of 0.5 Hz was applied for preprocessing, ensuring the removal of low-frequency noise.

#### 2.4.3. StO_2_ Signal Processing

The StO_2_ measured by the NIRS sensor was used to calculate the decrease in StO_2_ (ΔStO_2_) by subtracting the minimum value in the 3 min pressure interval from the average value of the 30 s before pressure was applied (Equation (2)) [30]. The average and standard deviation during the 3 min pressure interval were also extracted as StO_2_ features.(2)$$Decrease\;in\;StO_{2}\left(\%\right)=Baseline\;StO_{2}-Minimum\;StO_{2}$$

### 2.5. Statistics

The EDA, HRV, and StO_2_ features that were extracted were used for statistical analysis to explore the differences in pressure intensity (0, 10, 20, and 30 kPa) and pain levels (no pain, low, medium, high). The assumption of normality was evaluated using the Shapiro–Wilk test [39]. The data were not standardized, so non-parametric methods, including the Friedman test and Wilcoxon signed-rank test, were applied considering inter-participant variability [40,41,42]. All statistical analyses were performed in Python 3.8 using the pingouin library.

### 2.6. Regression

A multiple linear regression model was used to identify features that were not highlighted in the statistical analysis but were meaningful for predicting pressure and pain levels while also exploring the complex interactions between variables. The regression model was designed based on the results presented in Section 2.5 Statistics. The dependent and independent variables in the regression equation were considered significant if the *p*-value was <0.05. Additionally, the Variance Inflation Factor (VIF) was required to be below 10 to avoid multicollinearity [43]. The average RMSE of feature combinations was calculated using the 5-fold cross-validation method to evaluate model performance. The regression model was constructed using Python 3.8, Statsmodels 0.14.1, and Scikit-learn 1.3.2.

### 2.7. Machine Learning

A machine learning classification model was performed using Python 3.8 and the Scikit-learn package 1.3.2 to examine whether the measured biosignals, pressure intensity, and pain levels can be classified. Five types of classification models were used: SVM (Linear, Poly), Logistic Regression, Random Forest, MLP, and KNN. A total of 240 samples, derived from 20 participants under four different pressure intensities and pain levels, were used. The dataset was split into an 8:2 ratio for training and validation. The Synthetic Minority Oversampling Technique (SMOTE, k = 5) method was applied to address class imbalance [44]. Each model was optimized using 5-fold cross-validation with grid search to adjust the hyperparameters. The parameters used for optimization are listed in Table 2. Model performance was evaluated by calculating accuracy, sensitivity, specificity, and the harmonic mean (F_1_ score), with classification models implemented by combining one to a maximum of three features per sensor (Equations (3) and (4)).(3)$$Accuracy=\frac{TP+TN}{TN+FP+FN+TP},\;Sensitivity=\frac{TP}{FN+TP}$$(4)$$Specificity=\frac{TN}{TN+FP},\;F_{1}score=2\times \frac{Sensitivity\times Specificity}{Sensitivity+Specificity}$$

## 3. Results

The Spearman correlation analysis revealed distinct patterns among features within and across sensors (EDA, HRV, and StO_2_). Most EDA features showed strong positive correlations with each other (r > 0.7), except for Mean tonic, which exhibited weak correlations with other EDA features (r < 0.3). HRV features demonstrated weak correlations between time and frequency domains (r < 0.4), while features within the same domain showed strong correlations (r > 0.7). Most StO_2_ features had strong inter-feature correlations (r > 0.7), except for Max StO_2_. Inter-sensor correlations were generally weak, with EDA and HRV, as well as HRV and StO_2_, showing correlations below r = 0.3. Moderate correlations (r = 0.3–0.5) were observed between EDA and StO_2_, particularly in the calf region (up to r = 0.57). Regression and classification models incorporated up to three features, selecting only one from each sensor (e.g., EDA, HRV, and StO_2_) to reduce redundancy and avoid overfitting. Table 3, Figure 5 and Figure 6 showed significant changes in biosignal features according to pressure intensity (0, 10, 20, 30 kPa) and pain levels (no pain, low, moderate, high). Biosignal data were analyzed using non-parametric statistical methods, including the Friedman test and Wilcoxon signed-rank test with a significance level of *p* < 0.05. VAS significantly increased with pressure intensity across the thigh, the knee, and the calf, showing the highest values under 30 kPa pressure conditions in the order of calf, thigh, knee.

### 3.1. Statistical Analysis of EDA and StO_2_

The Spearman correlation analysis showed that most EDA features had a positive correlation with VAS (r > 0.4). Max Tonic demonstrated the strongest correlation (r ≥ 0.6), with the highest values observed in the calf, while Std Tonic exhibited a weaker correlation (r ≤ 0.3). Statistical analysis showed that most EDA features increased according to pressure intensity (0, 10, 20, 30 kPa) among the thigh, knee, and calf. Post-hoc analysis showed no significant differences between 20 kPa and 30 kPa in the thigh. Std Tonic significantly increased at 20 and 30 kPa compared to 0 kPa. Mean Tonic significantly increased at 10, 20, and 30 kPa compared to 0 kPa but showed no significant differences among 10, 20, and 30 kPa in the thigh and knee. Mean Tonic in the calf showed no significant differences between 20 kPa and 30 kPa, while Std Tonic in the knee did not exhibit significant differences across pressure intensity. Post-hoc analysis indicated that MTVSymp mean, MTVSymp std, and TVSymp showed no significant differences between 20 and 30 kPa in the thigh and knee. All EDA features increased according to pain levels (no pain, low, moderate, high) among the thigh, knee, and calf. Post-hoc analysis revealed that other EDA features, excluding Mean Phasic, Max Tonic, Mean Tonic, TVSymp Max, and MTVSymp Max, showed no significant differences between no pain and low pain. Mean Phasic and Mean Tonic increased compared to no pain but showed no differences among pain levels (low, moderate, high) in the thigh. In the knee, EDA features showed no significant differences between ‘low’ and ‘moderate’. The TVSymp component and MTVSymp component increased at ‘moderate’ and ‘high’ compared to baseline in the thigh, knee, and calf. EDA components showed the highest values at 30 kPa and ‘high’ in the calf. The Spearman correlation analysis showed that ΔStO_2_ and Std StO_2_ exhibited a very strong correlation with VAS (r ≥ 0.79). ΔStO_2_ and Std StO_2_ increased with higher pressure intensity and pain levels across all regions, while Mean StO_2_ decreased. The largest changes in StO_2_ occurred between 10 and 20 kPa and between low and moderate pain levels, with smaller changes observed between 20 and 30 kPa and between moderate and high pain levels. Max StO_2_ showed no significant differences according to pressure intensity and pain levels. Post-hoc analysis revealed that Mean StO_2_ showed no significant differences according to pressure intensity and pain levels in the knee. More detailed values were presented in Table A1 of Appendix A.

### 3.2. Statistical Analysis of HRV

Most HRV features showed weak correlations with VAS, among which HF exhibited the strongest correlation at r = −0.23. The Friedman test for pressure intensity (0, 10, 20, 30 kPa) revealed significant differences in HRV features across the thigh, knee, and calf. MaxNN, pNN20, HF, and HFn showed significant differences across these regions. Post-hoc analysis indicated that these features significantly increased at 10, 20, and 30 kPa compared to 0 kPa in the thigh, knee, and calf. However, no significant differences were observed between 20 kPa and 30 kPa in the thigh. In the calf, MaxNN exhibited significant differences across all pressure intensities, while no significant differences were observed among pressure intensities in the knee. pNN20 significantly increased in the thigh, knee, and calf compared to 0 kPa but showed no significant differences among pressure intensity in the knee. HF and HFn significantly decreased among the thigh, knee, and calf compared to 0 kPa; however, no significant differences were observed among pressure intensity (10, 20, 30 kPa). In the thigh and calf, additional HRV features, including MeanNN, MeanHR, MinHR, pNN50, and LF, exhibited significant differences. Post-hoc analysis revealed that MeanNN did not show significant differences among pressure intensity in the thigh, while it decreased from 10 kPa to 20 kPa in the calf. MeanHR at 30 kPa was smaller than at 0 kPa in the thigh, while in the calf, it decreased with increasing pressure intensity, except between 20 kPa and 30 kPa. MinHR decreased according to pressure intensity in the thigh and calf. Post-hoc analysis showed it was significantly smaller at 10, 20, and 30 kPa compared to 0 kPa only in the thigh. pNN50 increased with pressure intensity in the thigh and calf, but post-hoc analysis revealed significant increases only in the calf at 10, 20, and 30 kPa compared to 0 kPa. The Friedman test revealed significant differences in MaxNN and MinHR between no pain and pain levels (no pain, low, moderate, high) in the knee and calf. Post-hoc analysis showed that MaxNN significantly increased from ‘no pain’ to ‘high’ in the knee but significantly decreased in the calf under the same pain levels. HF significantly decreased across all pain levels (low, moderate, high) compared to ‘no pain’ in both the thigh and calf. The calf showed additional significant differences in MeanHR, pNN50, and pNN20. Post-hoc analysis revealed that MeanHR showed significant differences between ‘no pain’ and ‘pain levels (low, moderate, high)’, while pNN50 showed significant increases only between no pain and moderate pain levels. pNN20 did not exhibit significant differences across pain levels in post-hoc analysis. Other HRV features did not show significant differences among pain levels. More detailed values were presented in Table A1 of Appendix A.

### 3.3. Regression

Table 4 showed the results of the multiple linear regression model with the highest explanatory power (R^2^). Separate regression equations were constructed for each body part due to differences in biosignal patterns among the thigh, knee, and calf. The regression model results showed that pressure intensity prediction had a higher R^2^ value compared to VAS prediction in the thigh and knee, while the calf showed comparable results for both. The calf showed the highest explanatory power, with the pressure intensity prediction model showing R^2^ = 0.668 and RMSE = 6.516, and the VAS prediction model showing R^2^ = 0.668 and RMSE = 1.969. The prediction results showed a tendency to deviate further from the regression line as pressure intensity and pain levels increased.

### 3.4. Machine Learning Classification

Table 5 presented the machine learning classification results for pressure intensity and pain level. Pain level was divided into four categories based on VAS scores: no pain (VAS: 0), low (VAS: 1–3), moderate (VAS: 4–6), and high (VAS: 7–10). Pressure intensity was classified into four levels: 0, 10, 20, and 30 kPa. The pain level classification model showed lower accuracy with linear classifiers (SVM, LR) but performed better with nonlinear models (Random Forest, MLP, KNN). The RF (Random Forest) model achieved the highest accuracy for both pain level classification (88.2%) and pressure intensity classification (83.3%). Models using specific HRV features (e.g., RMSSD, SDNN, HF, LFHF) improved accuracy by 4–6%. Figure 7 shows the highest accuracy for both the pain level and pressure intensity classification models. Both models showed declining accuracy as pain levels and pressure intensity increased. The classification model of pressure intensity produced more correct classifications than misclassifications across all intensities, while in the pain level model, moderate levels were misclassified as high more often (67%) than they were correctly classified (17%).

## 4. Discussion

### 4.1. Pain and Biosignal

This study presents the results of quantifying pain using physiological signals under various pressure intensities applied to the thigh, knee, and calf. Most EDA features increased with higher pressure intensities and pain levels, showing a strong positive correlation with VAS and pressure. Among EDA components, the Max component significantly distinguished between pressure intensity and pain levels in post-hoc tests, aligning with previous studies [7,8,9,16,21] reporting similar findings for external pain stimuli. However, its distinguishing ability declined at higher pressure intensities and pain levels. These findings were consistent with previous studies [29,31,33,34], indicating that pressures above 20 kPa may cause discomfort or pain. Significant differences in MaxNN, pNN20, HF, and HFn were observed for different pressure intensities, and an increase in NN intervals reflected a relative reduction in heart rate. This response seemed to aim at mitigating pain and adapting to high-pressure intensity and pain levels [22,23,24]. Although NN intervals and HR differed significantly across pain levels, their patterns were inconsistent. HF decreased compared to baseline, aligning with previous studies [25,45,46], but did not show significant differences among pressure intensity and pain level. Previous studies [22,23,47] reported significant changes in LF, LFHF, RMSSD, and SDNN in response to pain. Similarly, this study found a significant reduction in LF during pressure stimulation in the thigh and calf, while in the thigh and knee, LFHF increased compared to the baseline. The reduction in HF can be interpreted as an inhibitory effect on the parasympathetic nervous system caused by initial cuff pressure, reflecting the body’s defensive response. As the pressure persisted, increases in NN intervals and pNN20 indicated the body’s compensatory responses and efforts to maintain autonomic balance. These changes suggest an adaptation process to the pressure stimulus, consistent with previous studies reporting pressure adaptation after 2–3 min of application [48]. However, some HRV features, including LFn for pressure intensity and MeanNN, MaxHR, RMSSD, and SDNN for pain levels, did not show significant differences across the three pressure intensities (10, 20, 30 kPa) or the three pain levels (low, moderate, high). These results aligned with previous research [45,49], which reported the difficulty of using single HRV parameters to distinguish pain levels. It also underscores the challenges of interpreting HRV compared to EDA. Compared to EDA, HRV showed weaker correlations and less distinct results regarding pressure intensity and pain levels, which may be attributed to the study’s focus on healthy adults. Previous studies [50,51] suggested that pressure pain activates different nerve fibers than thermal or electrical stimuli, leading to differing outcomes. The differences in neural responses observed in this study may also reflect the unique characteristics of pressure stimuli compared to thermal or electrical pain. HRV features are relatively challenging to use for differentiating pressure intensity and pain levels. They nonetheless play an essential role in understanding pain perception, as indicated by significant differences from baseline. MaxNN and HF emerged as notable features.

### 4.2. Differences Among the Thigh, Knee, and Calf

Differences in pain and physiological signal features were observed across the thigh, knee, and calf, even under the same pressure intensity. VAS scores reported by participants followed the order of calf > knee > thigh, with changes in EDA, HRV, and StO_2_ prominently observed in the calf, which resulted from differences in tissue composition [52,53]. The cuff was applied to the thigh, knee, and calf, as shown in Figure 3. Previous studies have also reported differences in pain perception across body regions [54]. The pain threshold (PTT) for each region was found to follow the order calf < knee < thigh, with the lowest PTT in the calf [55], which was consistent with the findings of the present study. The higher pain reported in the calf is likely due to the relative position of bones. While the thigh and knee have bones centrally located within the anatomical region influenced by cuff pressure, the tibia in the calf is positioned closer to the surface, leading to greater bulk deformation during pressure application [56,57]. Pressure applied to the calf induced greater pain compared to other regions, indicating the pronounced effects of cuff pressure in the calf when sitting. Variations in pain and discomfort perception across the thigh, knee, and calf may influence sympathetic nervous responses. Further study is necessary to better understand and evaluate pain differences among the thigh, knee, and calf.

### 4.3. Regression and Classification Model with BioSignal

The pressure intensity prediction model exhibited relatively higher explanatory power than the VAS prediction model, likely because VAS reflects the subjective evaluations of participants. The highest R^2^ values in this study were observed for the VAS prediction model and the pressure intensity prediction model, both at R^2^ = 0.668. Previous studies utilizing biosignals for regression analysis of pain levels and intensities reported lower explanatory power. For instance, previous studies using EDA features and VAS showed R^2^ = 0.477 for pain levels, R^2^ = 0.357 for pressure intensity, and R^2^ = 0.429 for pain intensity [8,19]. Other studies combining EDA and HRV features reported R^2^ = 0.24 [58]. In contrast to previous studies [8,19,20] that adjusted stimulus intensity according to participants’ perceived pain levels, the present study applied consistent pressure intensity of 10, 20, and 30 kPa across all participants to achieve more objective results.

This study achieved a maximum accuracy of 88.2% for four-level pain classification, surpassing previous research. For comparison, Posada-Quintero et al. [8] reported 56.9% for pain intensity and 51.6% for pain level classification using EDA alone in heat-induced pain; Lopez-Martinez et al. [58] reached 74.2% in a binary pain detection task; Kong et al. [19] obtained 63% for three-level classification of electrically induced pain; and Jiang et al. [47] achieved 83.3% for four-level classification of electrical and thermal pain. The improved accuracy in this study is partly attributed to incorporating StO_2_, which reflects blood flow information, combined with sympathetic nervous system responses. Linear function-based models (L-SVM, P-SVM, LR) were relatively more accurate in classifying pressure intensity, whereas non-linear models (Random Forest, MLP, KNN) showed higher accuracy in classifying pain levels. Non-linear models appear better suited for VAS-based pain level classification than linear models, likely because simple linear classifiers cannot fully capture the subjective nature of VAS evaluations. Further research should explore higher-dimensional aspects of pain perception, given this study’s limitations in fully elucidating the underlying mechanisms.

Table 6 shows the average accuracy of each biosignal feature in the Random Forest model for pain level classification. Results suggested that combining the right biosignal features can effectively quantify pain, with many three-feature combinations outperforming one- or two-feature sets. However, some combinations performed worse than individual features. For example, MTVSymp max and RMSSD achieved accuracy of 25% and 23% separately, but their combination dropped accuracy to 14%. These results align with previous findings [58], which indicate that combining biosignals does not always enhance model performance. Further research is needed to determine the optimal combination of biosignals. This study demonstrated greater explanatory power than prior work, highlighting the potential of biosignal-based pain quantification. Integrating various biosignals provided high explanatory power for predicting both pain and pressure intensity, consistent with previous studies [45] suggesting that combining multiple signals can enhance pain-related predictions. Future research should identify and compare effective biosignal combinations to improve pain quantification. This study demonstrated the effectiveness of combining sympathetic nervous system responses with blood flow characteristics to classify pressure intensity and pain. Such efforts lay the groundwork for biosignal-based pain quantification and hold potential for evaluating discomfort or pain in cuff-based robotic interfaces, as well as predicting and assessing real-time pain in medical procedures.

## 5. Limitation

The experiment was conducted in a seated position with subjects minimizing movement to observe sympathetic nervous activity and StO_2_. Biosignals are affected by movement, and different results can occur in dynamic conditions. Previous studies have reported differences in EDA and HRV signals according to age [26,27,42]. This experiment targeted healthy Asian adults in their 20s, but further experiments involving a wider range of age groups and diverse ethnicities are needed for practical applications. The NIRS may also be influenced by factors such as skin color and age [33,34,35,59], indicating the need for experiments that account for these variables. The applied pressure induced brief, repetitive episodes of pain rather than sustained discomfort or pain. The 3 min duration was chosen to prevent potential injury risks, such as muscle cramps or skin discoloration, that could arise from prolonged pressure. Long-term pressure may induce unintended sympathetic nervous responses, such as visual, auditory, and cognitive reactions, potentially complicating the interpretation of biosignals. This study specifically focused on quantifying discomfort and pain at the binding parts of wearable robots. However, this fixed duration may have limited studies to evaluate the effects of varying pressure durations on discomfort and pain. Future studies are needed to investigate both shorter and longer durations to achieve a more comprehensive understanding of these impacts. Changes in StO_2_ were more evident between 10 and 20 kPa and between low and medium pain levels. This suggests that further analysis in these ranges would be critical for quantifying discomfort and pain based on blood flow changes. Since the regression and classification models were based on features extracted during the 3 min pressure application, variations in pressure duration could affect model performance. Additional research on the effects of pressure duration within the 10–20 kPa range is also needed. The relationship between biosignals, VAS, and pain remains unclear [60]. Numerous studies underscore the importance of considering various factors, such as individual differences in pain perception [61,62], variability in biosignals, and gender- and age-related disparities in biosignals. The subjective nature of participant-reported VAS may have influenced the regression and machine learning models and should not be disregarded [63,64]. Further investigation is required to address the potential effects of individual variability in sympathetic nervous system responses, including pain tolerance and psychological factors. This study alone cannot fully elucidate the complex neural transmission processes linking perceived pain and biosignals. Future studies should include experiments with a larger and more diverse group of participants, such as patients or individuals with chronic conditions. This experiment was conducted in a controlled environment, which may not fully reflect real-world application settings where sympathetic nervous responses (EDA, HRV) could differ. Participants were kept in a comfortable state, but the potential influence of individual emotional states on sympathetic nervous responses remains a limitation of this study.

## 6. Conclusions

This study aims to quantify pain and discomfort induced by pressure. The correlation between EDA, HRV, StO_2_ changes, and VAS was analyzed in the thigh, knee, and calf under varying pressure intensities. Statistical analysis revealed significant changes in EDA, HRV, and StO_2_ in response to pressure intensity and pain level at the thigh, knee, and calf. VAS and pressure intensity regression models showed a maximum performance of R^2^ 0.668. The maximum accuracy for pain level classification was 88.2%, and for pressure intensity classification, it was 83.3%. The classification accuracy for pressure intensity was higher in the thigh and knee, while the calf showed better accuracy for pain level classification. EDA, HRV, and StO_2_ appear to be capable of quantifying user discomfort and pain. The study can contribute not only to quantifying pain but also to the design of human-machine interfaces and holds potential for real-time pain level classification.

## Figures and Tables

**Figure 1 sensors-25-00680-f001:**
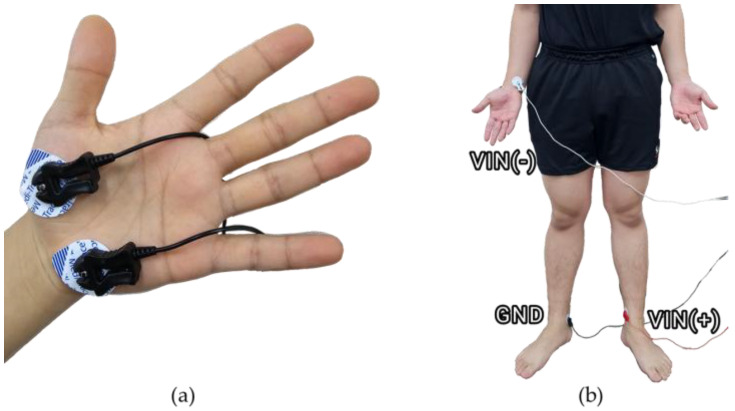
Equipment setting: (**a**) EDA sensor; (**b**) ECG sensor.

**Figure 2 sensors-25-00680-f002:**
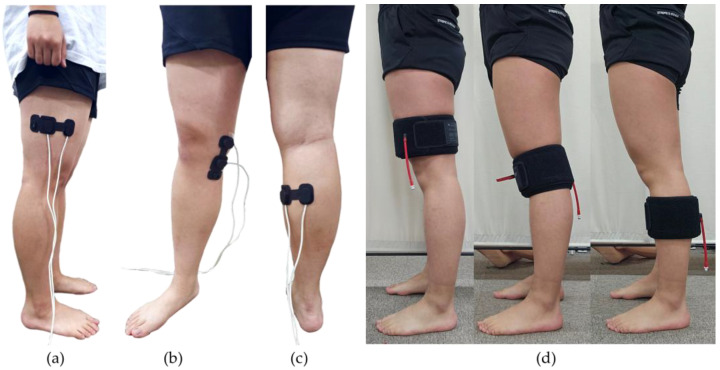
NIRS sensor positions: (**a**) Thigh; (**b**) Knee; (**c**) Calf; (**d**) Experimental setup.

**Figure 3 sensors-25-00680-f003:**
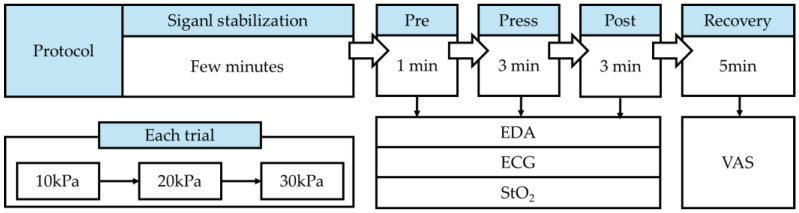
Experimental protocol.

**Figure 4 sensors-25-00680-f004:**
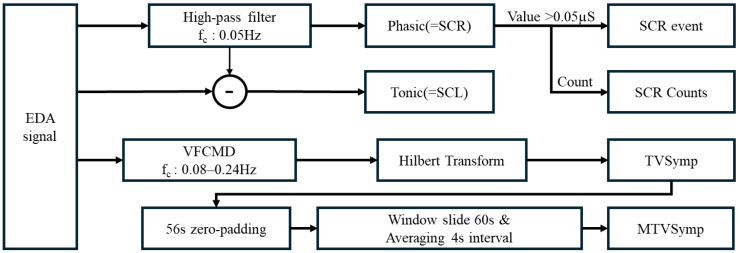
Signal processing of EDA.

**Figure 5 sensors-25-00680-f005:**
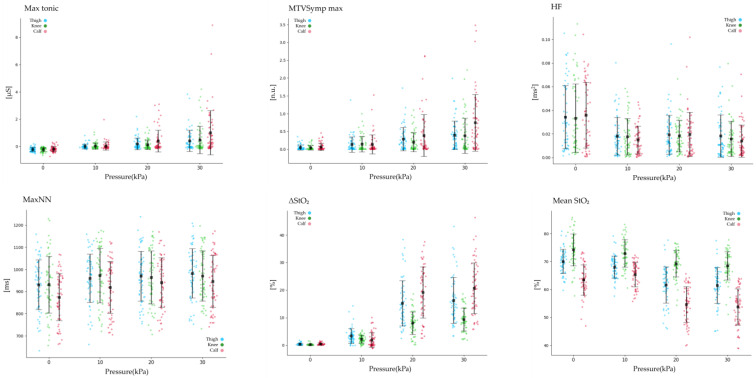
The significant results of biosignals for different pressure intensities. (Black dots: mean; lines: standard deviation).

**Figure 6 sensors-25-00680-f006:**
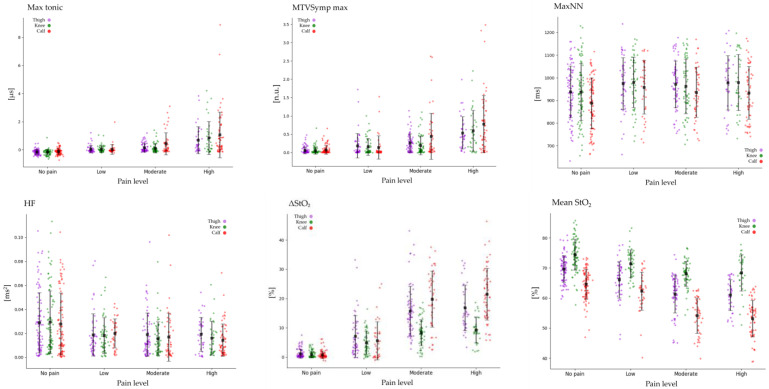
The significant results of biosignals for different pain levels. (Black dots: mean; lines: standard deviation).

**Figure 7 sensors-25-00680-f007:**
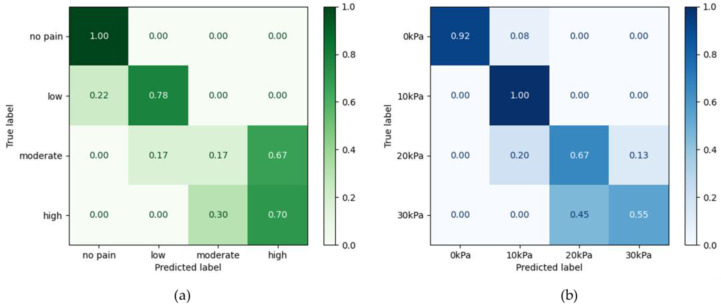
Confusion Matrix of the Highest Accuracy Model: (**a**) Pain level (**b**) Pressure intensity.

**Table 1 sensors-25-00680-t001:** NIRS sensor positions.

Thigh	The 2/3 position between the greater trochanter and the lateral epicondyle
Knee	The position between the medial epicondyle and medial condyle
Calf	The muscle belly of the medial gastrocnemius

**Table 2 sensors-25-00680-t002:** Parameter candidates for each classifier.

Classifiers	Parameters	Values
Support Vector Machine	C	1, 10, 100, 1000
	Gamma	0.0001, 0.001, 0.01, 0.1
	Degree (poly)	2, 3, 4
Logistic Regression	Solver	Newton-CG, LBFGS,
Random Forest	Criterion	Gini, Entropy
Multi-layer perceptron	Hidden Layer	1, 2, 3 (Hidden unit: 100)
	Activation	Logistic, tanh, relu
	Solvers	Adam, Stochastic gradient descent
	Learning rate	0.0001, 0.001, 0.01
K-nearest neighbors	K	3, 5, 7, 9

**Table 3 sensors-25-00680-t003:** The significant results of each biosignals features for different pressure intensity and pain level.

Features	Position	Pressure Intensity				Pain Level			
		0 kPa	10 kPa	20 kPa	30 kPa	No Pain	Low	Moderate	High
Max Tonic [μS]	Thigh	−0.22 ± 0.15	0 ± 0.1$8^{0}$	0.2 ± 0.4$3^{0,1}$	0.41 ± 0.7$8^{0,1}$	−0.16 ± 0.18	0.04 ± 0.2$7^{0}$	0.15 ± 0.$3^{0,1}$	0.68 ± 0.9$6^{0,1}$
	Knee	−0.22 ± 0.15	0.03 ± 0.2$3_{a}^{0}$	0.13 ± 0.3$5^{0,1}$	0.48 ± $1_{a}^{0,1}$	−0.17 ± 0.18	0.06 ± 0.2$5^{0}$	0.1 ± 0.3$2_{a}^{0}$	0.84 ± 1.1$8^{0,1}$
	Calf	−0.19 ± 0.19	0.03 ± 0.$3_{a,b}^{0}$	0.4 ± 0.8$1_{b}^{0,1}$	1.01 ± 1.6$2_{a,b}^{0,1,2}$	−0.12 ± 0.21	0.03 ± 0.3$5^{0}$	0.43 ± 0.7$9_{a,b}^{0}$	1.07 ± 1.6$3^{0,1,2}$
MTVSymp max [n.u.]	Thigh	0.05 ± 0.07	0.14 ± 0.2$2^{0}$	0.29 ± 0.3$3^{0,1}$	0.39 ± 0.$4^{0,1}$	0.06 ± 0.09	0.19 ± 0.3$3^{0}$	0.27 ± 0.2$5^{0}$	0.54 ± 0.4$4^{0,1,2}$
	Knee	0.04 ± 0.06	0.14 ± 0.22	0.2 ± 0.2$7^{0}$	0.38 ± 0.$5^{0,1}$	0.05 ± 0.0$9_{a}$	0.16 ± 0.23	0.19 ± 0.2$5^{0}$	0.6 ± 0.5$6^{0,1}$
	Calf	0.07 ± 0.1	0.14 ± 0.27	0.39 ± 0.5$9^{0,1}$	0.75 ± 0.7$9_{a,b}^{0,1,2}$	0.08 ± 0.1$2_{a,b}$	0.14 ± 0.31	0.44 ± 0.6$3_{a,b}^{0,1}$	0.78 ± 0.7$8^{0,1}$
MaxNN [ms]	Thigh	931 ± 112.4	961 ± 109.$2^{0}$	970 ± 112.$7^{0}$	982 ± 111.$4^{0,1}$	938.2 ± 113	974.2 ± 114.3	971.4 ± 103	978.4 ± 120
	Knee	932 ± 128.0	973 ± 122.$1^{0}$	964 ± 119.$4^{0}$	971 ± 113.$1^{0}$	937.5 ± 126	980.2 ± 111	962.4 ± 120	980.1 ± 12$3^{0}$
	Calf	874.3 ± 10$6_{a,b}$	919.2 ± 11$7_{a,b}^{0}$	940.6 ± 11$2_{a,b}^{0,1}$	946.3 ± 11$7_{a,b}^{0,1}$	937.8 ± 112	958.3 ± 117.3	936 ± 110.1	933.7 ± 11$7_{a,b}^{0}$
HF [ms^2^]	Thigh	0.034 ± 0.027	0.018 ± 0.01$6^{0}$	0.019 ± 0.01$6^{0}$	0.018 ± 0.01$8^{0}$	0.029 ± 0.025	0.019 ± 0.01$8^{0}$	0.019 ± 0.01$8^{0}$	0.019 ± 0.01$4^{0}$
	Knee	0.033 ± 0.029	0.018 ± 0.01$5^{0}$	0.018 ± 0.01$3^{0}$	0.016 ± 0.01$5^{0}$	0.028 ± 0.02$6_{a}$	0.02 ± 0.012	0.017 ± 0.02	0.014 ± 0.014
	Calf	0.036 ± 0.028	0.015 ± 0.01$2^{0}$	0.02 ± 0.01$9^{0}$	0.014 ± 0.01$4^{0}$	0.028 ± 0.02$3_{a,b}$	0.02 ± 0.01$2^{0}$	0.016 ± 0.0$2^{0}$	0.014 ± 0.01$4^{0}$
ΔStO_2_ [%]	Thigh	0.39 ± 0.34	3.36 ± 2.$7^{0}$	15.24 ± 8.1$5^{0,1}$	16.3 ± 8.3$1^{0,1,2}$	1.15 ± 1.54	7.09 ± 7.0$8^{0}$	15.75 ± 8.4$5^{0,1}$	16.87 ± 7.7$9^{0,1}$
	Knee	0.24 ± 0.2$8_{a}$	2.19 ± 1.3$7^{0}$	8.05 ± 4.0$9_{a}^{0,1}$	9.41 ± 4.1$5_{a}^{0,1,2}$	0.87 ± 1.14	4.93 ± 3.9$3_{a}^{0}$	8.34 ± 4.4$6_{a}^{0,1}$	9.29 ± 4.4$1_{a}^{0,1,2}$
	Calf	0.47 ± 0.3$7_{b}$	1.89 ± 2.5$9_{b}^{0}$	19.19 ± 9.2$8_{a,b}^{0,1}$	20.68 ± 9.$2_{a,b}^{0,1,2}$	0.81 ± 1.26	5.65 ± 6.4$7^{0}$	19.8 ± 9.5$7_{a,b}^{0,1}$	21.53 ± 8.7$9_{a,b}^{0,1}$
Std StO_2_ [%]	Thigh	0.23 ± 0.19	0.89 ± 0.6$8^{0}$	4.36 ± 2.7$1^{0,1}$	4.65 ± 2.8$3^{0,1,2}$	0.4 ± 0.39	2.01 ± 2.3$4^{0}$	4.51 ± 2.8$5^{0,1}$	4.73 ± 2.$5^{0,1}$
	Knee	0.13 ± 0.12	0.55 ± 0.2$9^{0}$	1.81 ± 0.9$8^{0,1}$	2.07 ± 1.0$2^{0,1,2}$	0.24 ± 0.21	1.11 ± 0.7$4^{0}$	1.91 ± 1.0$8^{0,1}$	2.04 ± 1.1$2^{0,1}$
	Calf	0.27 ± 0.2$3_{a,b}$	1.29 ± 0.7$1_{a,b}^{0}$	6.36 ± 2.7$5_{a,b}^{0,1}$	6.72 ± 2.$9_{a,b}^{0,1}$	0.52 ± 0.4$5_{a,b}$	2.57 ± 1.8$9_{a,b}^{0}$	6.49 ± 2.7$5_{a,b}^{0,1}$	6.99 ± 2.8$3_{a,b}^{0,1}$
VAS	Thigh	0 ± 0	0.85 ± 1.0$1^{0}$	4.23 ± 1.4$1^{0,1}$	6.35 ± 1.8$6^{0,1,2}$	0 ± 0	1.86 ± 0.8$8^{0}$	4.66 ± 0.8$3^{0,1}$	7.79 ± 0.7$4^{0,1,2}$
	Knee	0 ± 0	0.86 ± $1^{0}$	4.25 ± 1.2$9^{0,1}$	6.71 ± 1.7$7_{a}^{0,1,2}$	0 ± 0	2.04 ± 0.8$7^{0}$	5.06 ± 0.7$8^{0,1}$	7.99 ± 0.8$0^{0,1,2}$
	Calf	0 ± 0	0.65 ± 0.$9^{0}$	4.72 ± 1.$3^{0,1}$	7.94 ± 1.4$9_{a,b}^{0,1,2}$	0 ± 0	1.85 ± 0.9$1^{0}$	4.89 ± 0.7$7^{0,1}$	8.22 ± 0.8$0^{0,1,2}$

Superscript numbers indicate significant differences relative to specific pressure intensity or pain levels. Pressure intensity: 0 = 0 kPa, 1 = 10 kPa, 2 = 20 kPa; Pain level: 0 = no pain, 1 = low, 2 = moderate. Subscripts indicate significant differences in statistical tests among the thigh, knee, and calf (a = thigh, b = knee). Differences in features by Thigh, Knee, and Calf were assessed using the Friedman test (*p* < 0.05).

**Table 4 sensors-25-00680-t004:** The regression model with the highest Adjusted R^2^ for thigh, knee, and calf.

**Regression Model: VAS Prediction**			
**Body Part**	**R^2^**	**RMSE**	**Regression Equation**
Thigh	0.554	1.925	VAS = −1.969 + (1.647 × Max Tonic) + (0.042 × MeanHR) + (0.186 × ΔStO_2_)
Knee	0.643	1.798	VAS = 1.616 + (13.943 × TVSymp mean) + (−2.519 × HFn) + (0.394 × ΔStO_2_)
Calf	0.668	1.969	VAS = 1.001 + (0.121 × SCR counts) + (−14.585 × HF) + (0.211 × ΔStO_2_
**Regression Model: Pressure Prediction**			
Thigh	0.597	7.121	Pressure = 29.856 + (21.487 × Mean Tonic) + (−0.025 × MinNN) + (0.784 × ΔStO_2_)
Knee	0.687	6.245	Pressure = 8.714 + (1238.511 × Mean phasic) + (−8.118 × HFn) + (1.758 × Δ StO_2_)
Calf	0.668	6.516	Pressure = 11.409 + (15.880 × Mean Tonic) + (−70.098 × HF) + (2.081 × Std StO_2_)

**Table 5 sensors-25-00680-t005:** Machine learning classification results: accuracy for pain level and pressure intensity.

Classifier	Pain Level			Pressure Intensity		
	Thigh	Knee	Calf	Thigh	Knee	Calf
L-SVM	0.657	0.690	0.776	0.729	0.778	0.780
P-SVM	0.629	0.704	0.713	0.720	0.750	0.750
LR	0.657	0.690	0.776	0.708	0.729	0.783
RF	0.757	0.803	0.882	0.729	0.778	0.833
MLP	0.800	0.746	0.881	0.708	0.750	0.813
KNN	0.714	0.803	0.803	0.708	0.729	0.750

L-SVM: linear support vector machine; P-SVM: Polynomial Support Vector Machine. LR: Logistic regression, RF: Random Forest, MLP: Multi-layer perceptron, KNN: k-nearest neighbor. The model with the highest accuracy was presented for each category, and in the case of equal accuracy, the model with the higher F_1_ score was selected.

**Table 6 sensors-25-00680-t006:** Average Accuracy Based on Feature Combinations for the Best-Performing Model.

Feature Set (Count of Combinations)	Mean Accuracy ± Standard Deviation		
	Thigh	Knee	Calf
EDA (16)	0.397 ± 0.076	0.364 ± 0.071	0.425 ± 0.059
HRV (15)	0.353 ± 0.054	0.324 ± 0.050	0.346 ± 0.062
StO_2_ (4)	0.439 ± 0.086	0.500 ± 0.147	0.523 ± 0.136
EDA+HRV (240)	0.495 ± 0.058	0.468 ± 0.055	0.516 ± 0.054
EDA+StO_2_ (60)	0.545 ± 0.082	0.587 ± 0.084	0.660 ± 0.071
HRV+StO_2_ (51)	0.550 ± 0.104	0.567 ± 0.090	0.602 ± 0.095
EDA+HRV+StO_2_ (767)	0.599 ± 0.072	0.632 ± 0.076	0.693 ± 0.075

## Data Availability

The data presented in this study are available upon request from the corresponding author. The data are not publicly available because the authors are continuing the study.

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
