# Peer review of "Estimation of Pressure Pain in the Lower Limbs Using Electrodermal Activity, Tissue Oxygen Saturation, and Heart Rate Variability"

_sensors, 2025, doi:10.3390/s25030680_

Round 1
Reviewer 1 Report
Comments and Suggestions for Authors
The article investigates the quantification of pressure-induced pain and discomfort using physiological signals such as electrodermal activity (EDA), tissue oxygen saturation (StOâ‚‚), and heart rate variability (HRV). It employs statistical and machine learning methods to analyze the correlation of these signals with pain levels and pressure intensities across different lower limb regions. The findings indicate significant changes in EDA and StOâ‚‚ features with varying pressure intensities and pain levels, while HRV features showed limited significance. Regression and classification models demonstrated moderate accuracy, highlighting the potential of integrating bio-signals for objective pain assessment. I have the following major concerns regarding the design of this article:
1. The pressure durations were fixed at 3 minutes, which may not capture long-term discomfort or sustained pain effects. The study could benefit from varying pressure durations to analyze both short-term and long-term impacts on bio-signals.
2. While HRV features were analyzed, their limited significance in distinguishing between pressure intensities and pain levels raises questions about their utility. The authors should explore additional HRV metrics or advanced signal processing techniques to enhance their relevance.
3. The study’s machine learning models demonstrated moderate classification accuracy, particularly for pressure intensity. Incorporating additional bio-signals or refining feature selection could improve model performance.
4. The correlation analysis indicates weaker associations for certain features (e.g., SCR counts and tonic mean). The authors should provide a detailed discussion on why these features exhibited lower correlations and whether alternative methods could enhance their predictive capability.
5. The influence of skin tone on NIRS measurements was acknowledged but not rigorously addressed in the experiments. Including a diverse participant pool with varying skin tones could validate the robustness of StOâ‚‚ measurements.
6. The manuscript does not provide sufficient detail about the preprocessing techniques used for data standardization or handling noise in the bio-signal data. A more comprehensive explanation would enhance reproducibility and reliability.
7. Statistical tests revealed significant differences among regions (thigh, knee, and calf), but the discussion lacks a clear interpretation of why these differences occur. Exploring biomechanical or anatomical factors contributing to these variations would add depth to the findings.
8. The regression models, while achieving moderate R² values, do not provide a strong predictive capability. The authors should investigate more advanced regression techniques or nonlinear models to capture complex interactions among bio-signals.
9. The limitations section mentions several factors, but the discussion could be expanded to include the potential effects of inter-individual variability, such as pain tolerance and psychological factors, on the results.
Reviewer 2 Report
Comments and Suggestions for Authors
This paper presents a significant and thought-provoking research topic related to estimating the pain induced by pressure considering physiological signals such as changes in electrodermal activity, tissue oxygen saturation, heart rate variability, and Visual Analog Scale. Pressures of 10, 20, and 30 kPa were applied for 3 minutes to the thigh, knee, and calf to 20 healthy adults seated on a chair. The original contribution of this research should be clearly highlighted in the Introduction. As I understood, the calf is the most sensitive body area since the participants in the study reported the highest level of pain perceived by pressure applied to the calf. Also, the same authors published a previous paper entitled Measurements of Electrodermal Activity, Tissue Oxygen Saturation, and Visual Analog Scale for Different Cuff Pressures. Are the experimental results from the current and previous papers correlated from any point of view (similarities, differences, other insights, or useful observations)? Regarding the differences shown by black bars in the graphical representation, I recommend adding some details in the content/details of the paper. Regarding future direction, would you consider monitoring brainwaves using a portable EEG headset and making some investigations upon EEG rhythms depending on the level of pain? Regarding limitations of the proposed experimental procedure, you might include the influence of emotional states and environmental conditions (temperature) upon EDA signals. The robustness of the study is emphasized by the statistical analyses (RM ANOVA, Friedman tests, regression) and machine learning. Regarding the leveraging of the obtained experimental results, I wonder if would it be possible to create an automatic solution working in real-time for detecting the level of pain perceived by a patient during a medical procedure?
Reviewer 3 Report
Comments and Suggestions for Authors
This paper presents a novel approach to quantifying pain induced by pressure, integrating electrodermal activity (EDA), tissue oxygen saturation (StOâ‚‚), and heart rate variability (HRV) with machine learning. The use of machine learning models to analyze multiple physiological parameters can help to deeper understand of the link between physiological signals and pain. The research has certain innovation and application value.
However, there exist some questions surrounding this work.
1. The number of data set for the training and test utilized in machine learning is not mentioned in the text. The sample size of 20 volunteers might be insufficient to meet the requirements of some machine learning methods. To enhance the performance of the model, it might be needed to increase the sample size or select machine learning algorithms more suitable for small samples.
2. If the input feature for the classification algorithms covered all the features, is there any redundant feature affecting classification accuracy? As five classification algorithms were tested in the article, can the authors explain the relationship between the algorithm selection and the signal features, as well as the existence of redundant statistical data features and their influence on the classification results?
3. Given that both EDA and HRV are associated with sympathetic nervous system activity, it is curious that the HRV features did not show significant changes. It is recommended that the authors explore whether this lack of significant change in HRV features is related to the pressure intensity or the different physiological mechanisms, such as the short-term pain and the long-term pain.
4. The HRV features in the article did not show significant variations in relation to the intensity of pressure or the level of pain, which is inconsistent with the findings of some other studies. Can the authors analysis of the HRV results to clarify the aforementioned issues? Simultaneously, whether the reason why HRV can improve the accuracy of the model in machine learning classification is the combined effect of multiple HRV parameters can also be further analyzed.
5. The VAS scale relies on subjective responses. Does the author's choice of it as the standard have proofs and could it lead to the classification results being influenced by subjective evaluations?
6. Please verify the tables and figures in the text. For example, in Table 5, RF is marked as LF.
Round 2
Reviewer 1 Report
Comments and Suggestions for Authors
I am happy to see the massive amount of modifications the authors did in response to my earlier comments. The paper seems scientifically more sound and would be attracted by a wide array of audience. I am satisfied with the current version of the paper and recommend it for prospective acceptance. Best of luck to the authors.